# A Low-Noise Interface ASIC for MEMS Disk Resonator Gyroscope

**DOI:** 10.3390/mi14061256

**Published:** 2023-06-15

**Authors:** Wenbo Zhang, Liang Yin, Yihang Wang, Risheng Lv, Haifeng Zhang, Weiping Chen, Xiaowei Liu, Qiang Fu

**Affiliations:** 1MEMS Center, Harbin Institute of Technology, Harbin 150001, China; 17b321006@stu.hit.edu.cn (W.Z.); yinliang2003@126.com (L.Y.); 17b921023@stu.hit.edu.cn (Y.W.); zhanghf@hit.edu.cn (H.Z.); chenwp@hit.edu.cn (W.C.); lxw@hit.edu.cn (X.L.); 2Key Laboratory of Micro-Systems and Micro-Structures Manufacturing, Harbin Institute of Technology, Ministry of Education, Harbin 150001, China; 3China Academy of Information and Communications Technology, Beijing 100191, China; lvrisheng@caict.ac.cn; 4State Key Laboratory of Urban Water Resource & Environment, Harbin Institute of Technology, Harbin 150001, China

**Keywords:** ASIC, MEMS disk resonator gyroscope, force-to-rebalance, angle random walk, bias instability

## Abstract

This paper proposes a low-noise interface application-specific integrated circuit (ASIC) for a microelectromechanical systems (MEMS) disk resonator gyroscope (DRG) which operates in force-to-rebalance (FTR) mode. The ASIC employs an analog closed-loop control scheme which incorporates a self-excited drive loop, a rate loop and a quadrature loop. A ΣΔ modulator and a digital filter are also contained in the design to digitize the analog output besides the control loops. The clocks for the modulator and digital circuits are both generated by the self-clocking circuit, which avoids the requirement of additional quartz crystal. A system-level noise model is established to determine the contribution of each noise source in order to reduce the noise at the output. A noise optimization solution suitable for chip integration is proposed based on system-level analysis, which can effectively avoid the effects of the 1/f noise of the PI amplifier and the white noise of the feedback element. A performance of 0.0075°/√h angle random walk (ARW) and 0.038°/h bias instability (BI) is achieved using the proposed noise optimization method. The ASIC is fabricated in a 0.35 μm process with a die area of 4.4 mm × 4.5 mm and power consumption of 50 mW.

## 1. Introduction

Planar microelectromechanical systems (MEMS) gyroscopes have outstanding advantages in terms of size, power consumption and cost and are now widely used in the consumer market and low- and mid-range industrial markets based on micromechanical techniques [1,2,3]. Several kinds of MEMS gyroscopes have reached near-navigation accuracy in recent years with the advances in fabrication processes and the optimization of gyroscope structures [4,5,6,7,8]. Among these MEMS gyroscopes, the disk resonator gyroscope (DRG) shows excellent potential to be the next-generation MEMS gyroscope due to its fully symmetrical structure and perfect degenerate working modes [4,5]. Nowadays, the interface circuit of the DRG is mainly realized by discrete devices, which results in high power consumption and large size and seriously restricts the advantages of MEMS devices themselves. Therefore, research on interface ASICs for MEMS DRGs has great significance for their practical implementation.

Proper selection of the operation mode and circuit control scheme is the precondition for chip integration. The main operation modes of an MEMS DRG include full-angle mode and force-to-rebalance (FTR) mode. The full-angle mode, also known as the rate-integrating mode, offers significant advantages in terms of measurement range and bandwidth. However, the performance under full-angle mode is highly susceptible to processing errors [9,10]. In FTR mode, the MEMS DRG is less affected by the process, and it easy to achieve higher accuracy [5]. Therefore, the FTR mode is selected here to achieve higher accuracy.

The circuit control methods for MEMS DRGs are mainly divided into the analog scheme and the mixed digital–analog scheme. In the analog scheme, the sensing and processing of the signals are both implemented by analog circuits, and the structure is relatively simple. In mixed digital–analog solutions, the sensing and feedback of the signals are implemented in analog circuits, while the processing is implemented in digital circuits. The advantage of this scheme is that the signal processing is more flexible and easier to debug. However, this solution imposes high requirements on the data converters (ADCs, DACs) since the converters need a relatively larger bandwidth to process the gyroscope resonant signal. In additional, the typical digital–analog scheme requires two ADCs and three DACs [5], which occupy a large chip area. Therefore, this paper prefers the analog circuit scheme for system integration for the sake of compromise between detection accuracy, chip area, system power consumption and implementation cost.

High-precision detection is one of the main focuses of interface circuit design, and the output noise is an important factor which limits the detection accuracy. The contribution of noise sources in FTR gyroscope systems has been tested and discussed in some studies [11,12]; however, the gyroscope circuits in these articles are built by discrete devices and unable to be optimized specifically. Therefore, traditional discrete device solutions typically involve improving the accuracy by purchasing expensive devices, which undoubtedly increases system costs.

Integrated circuits allow the design of circuit elements to be optimized for specificity and can achieve a compromise between power consumption, area, cost and performance. The main contribution of this paper is the low-noise interface ASIC design for the complex MEMS DRG system. This paper gives a detailed analysis of the noise contribution of each circuit element in the system, based on which a noise optimization solution suitable for chip integration is proposed. The noise optimization method proposed in this paper can effectively avoid the effects of the 1/f noise of the PI amplifier and the white noise of the feedback element. A performance of 0.0075°/√h angle random walk (ARW) and 0.038°/h bias instability (BI) is achieved by using the proposed method. The ASIC also incorporates a ΣD modulator and a digital circuit for digital output.

This paper is organized as follows: Section 2 describes the MEMS DRG we used and the system architecture we proposed. Section 3 focuses on the effect of each noise source on the output under FTR mode. Section 4 gives the implementation details. The experimental results are presented and discussed in Section 5, and the paper ends with the conclusions given in Section 6.

## 2. System Overview

### 2.1. MEMS DRG Structure

The gyroscope system consists of a MEMS sensing element and an interface ASIC, and the characteristics of the sensing element directly determine the performance. The structure and key parameters of the MEMS DRG we used are shown in Figure 1 and Table 1, respectively. The following advantages ensure the performance of the MEMS DRG.

#### 2.1.1. High-Quality Factor (Q)

The ARW caused by mechanical-thermal noise in FTR mode can be computed by the following [13]:(1)$$\Omega_{r\omega }\approx \sqrt{\frac{k_{B}T\omega_{y}}{A_{x}^{2}m\omega_{x}^{2}Q_{y}}\left(1+\frac{\Delta \omega^{2}}{\omega_{m}^{2}}\right)}\times 3437.7{}^{\circ}/\sqrt{h}$$
where *k_B_* is Boltzmann’s constant; *T* is the temperature in degrees of Kelvin; *A_x_* and *m* are the oscillation amplitude along drive mode and the effective mass of the gyroscope, respectively; *ω_x/y_* and *Q_x/y_* are the resonance frequencies and quality factors of drive and sense mode, respectively; and Δ*ω* and *ω_m_* represent the frequency split and mechanical bandwidth of the gyroscope, respectively. Equation (1) shows that the high-quality factor, which determines the white noise contribution in the MEMS sensing element, is the essential attribute of a good performance for mode-matched gyroscopes. The *Q* factor of an optimized DRG prototype reaches 650 k by using *Q* factor improvement methods such as spoke length distribution optimization and lumped mass configuration, which guarantee the AWR of the sensor itself lower than 0.001°/√h [5,14].

#### 2.1.2. Large Oscillation Amplitude

According to Equation (1), increasing the oscillation amplitude *A_x_* is an obvious way to reduce mechanical-thermal noise. However, the maximum amplitude may be limited by nonlinear problems. The electrostatic nonlinearity and capacitive nonlinearity can be reduced and the ultimate oscillation amplitude can be increased to 49% of the capacitive gap by optimizing the electrode arrangement based on the vibration amplification [15].

#### 2.1.3. Fully Symmetrical Structure

A fully symmetrical structure ensures the parameters of the DRG changing synchronously when the environment changes. The MEMS DRG used in this design has high immunity to fabrication error due to the in-plane isotropic properties of the <111> crystal orientation and the optimized structure [16]; thus, the frequency split Δ*ω* varies slightly as the temperature changes. This advantage alleviates the need for a frequency control loop and can avoid the noise contribution on BI caused by frequency control circuits [11].

#### 2.1.4. Abundant Electrodes

The electrode configuration of the DRG is shown in Figure 1a. The DRG contains both outer and inner electrodes, which are divided as sixteen separate electrodes. The electrodes in the 0° (90°, 180°, 270°) and 45° (135°, 225°, 315°) directions are drive/sense electrodes. The electrodes of these directions can also act as frequency tuning electrodes. The inner and outer electrodes in the 22.5° (112.5°, 202.5°, 292.5°) and 67.5° (157.5°, 247.5°, 337.5°) directions are quadrature tuning electrodes. In the circuit testing, one end is connected to the ground of the PCB board and the other to the quadrature feedback voltage. Figure 1b shows the inner electrodes in driving and sensing directions. The two electrodes in the same mesh are divided into a positive and negative electrode, which is separated by an isolation layer in the middle. This configuration achieves differential driving and eliminates common mode error. In addition, the abundant inner electrodes could increase the capacitance sensibility and improve the efficiency of electromechanical conversion, which contribute to reducing output noise and tunning voltage. The quadrature tunning voltage is usually within 2 V.

### 2.2. MEMS DRG System Architecture

Figure 2 gives a detailed block diagram of the DRG system. The control circuit consists of a drive loop, a rate control circuit, a quadrature nulling loop, a self-clocking circuit, a low-pass ΣD modulator and a digital decimation filter.

The drive loop adopts the noise self-excited scheme. The front end of the drive circuit is a transimpedance amplifier (TIA), the output of which is quadrature with the resonator displacement. This signal is amplified by a VGA and applied to the drive actuation electrode (drvact±). Since the phase relationship between the actuation force and displacement at the resonant frequency is 90°, the phase shift of the drive loop is zero. According to the Barkhausen criterion, when the loop gain is larger than 1, positive feedback is established and circuit noise makes the loop self-excited at resonant frequency. The advantage of the self-excited start-up scheme is that no extra phase control loop is required in the loop. Once the resonator oscillates, the phase-locked loop (PLL) can be used to generate the various clocks required by the system. The VGA works at maximum gain for fast start-up during the initial operation of the drive circuit. The PI controller plays the role of maintaining the amplitude constant when the amplitude reaches the reference level.

The self-clocked circuit contains a comparator and the PLL, which is used for frequency locking, 90° phase shifting and frequency multiplication of the drive signal. In Figure 2, clk_x/v represent the output clocks in-phase with driving displacement and driving velocity, respectively, and they are used as modulation and demodulation clocks in the FTR loop. Additionally, clk_sys is a multiplier clock of clk_x, which is used for low-pass ΣΔ modulator and digital circuits.

The rate control loop and quadrature nulling loop ensure that the amplitude of the resonator in the sensing direction is zero, which guarantees that the displacement towards the sensing axis is in the FTR state. The output of the PI controller in the rate loop is the analog output of the DRG, and this signal is modulated and applied to the sensing actuation electrodes (snsact±). The DC quadrature nulling loop is used here, which benefits from the presence of quadrature electrodes in the MEMS DRG, as can be seen in Figure 2. The output of the PI controller (approximate DC signal) in the quadrature loop acts directly on the quadrature tuning electrodes to suppress the quadrature component. Since the voltage acting on the sensing actuation electrode is a resonant frequency signal, no electrical coupling exists from the quadrature electrode to the sensing electrode. In addition, the quadrature component before the PI controller is suppressed to zero due to the PI controller and thus cannot affect the rate sensing. Moreover, the DC quadrature nulling loop possesses better environmental adaption compared to the AC quadrature nulling method, which helps the gyroscope achieve better stability [17].

The ASIC also integrates the ΣΔ modulator and the digital filter to digitize the analog output. The gyroscope output signal is commonly at low frequency, typically within a few hundred Hertz. Therefore, a low-speed, high-precision ΣΔ ADC structure is implemented here. The digital circuit converts the one-bit stream signal from the output of the ΣΔ modulator to a digital signal through decimating and filtering.

The DC voltage added on the mass is 10 V, which is applied off-chip.

## 3. Noise Analysis of MEMS DRG System

A MEMS DRG system operating in FTR mode has a more complex structure and additional noise sources than conventional open-loop gyroscopes, as can be seen from the analysis in Section 2. Each noise source has a different impact on the output, so a system-level noise model is required to analyze the impacts of each source and to optimize them specifically. In addition, since the FTR system contains demodulation and modulation circuits, the signal frequencies processed by each unit are not the same. Therefore, the signals before and after modulation/demodulation need to be unified to make the analysis easier. In this section, the dynamic characteristics of the resonators under the sense mode are equated using the stochastic averaging method [18], and the influence of each noise source is analyzed based on it.

### 3.1. Slow Signal Equivalence of the DRG Dynamics

The stochastic averaging method is used to simplify the dynamic model of the resonant oscillator. This method extracts information about the slow signals (amplitude signals) and allows for a unified noise model analysis. Consider the typical gyroscopic dynamical model:(2)$$\ddot{x}+\frac{\omega_{x}}{Q_{x}}\dot{x}+\omega_{x}^{2}x-2nA_{g}\Omega \dot{y}=\frac{F_{d}}{m}\sin\left(\omega_{x}t\right)$$
(3)$$\ddot{y}+\frac{\omega_{y}}{Q_{y}}\dot{y}+\omega_{y}^{2}y+2nA_{g}\Omega \dot{x}=\frac{F_{s}}{m}\sin\left(\omega_{x}t\right)$$
where *F_d_* and *F_s_* are the amplitudes of driving force and sensing feedback force, respectively; *n* = 2 is the wineglass mode and *A_g_* = 0.4 is the angular gain for the MEMS DRG. Let $u=\dot{x}/\omega_{x}$, $v=\dot{y}/\omega_{x}$; the second-order dynamics model of the gyroscope can be rewritten as follows:(4)$$\dot{x}=\omega_{x}u$$
(5)$$\dot{u}=-\omega_{x}x-\frac{\omega_{x}}{Q_{x}}u+\frac{F_{d}\sin\left(\omega_{x}t\right)}{m\omega_{x}}+2nA_{g}\Omega v$$
(6)$$\dot{y}=\omega_{x}v$$
(7)$$\dot{v}=-\omega_{x}y-\frac{\omega_{y}}{Q_{y}}v+\frac{F_{s}}{m}\sin\left(\omega_{x}t\right)-2\Delta \omega y-2nA_{g}\Omega u$$

Since the drive circuit employs a self-excitation scheme, the drive force is 90° in phase with the resonator displacement when the amplitude is stable. However, the phase relationship between the feedback force and the displacement at the sensing axis is not perfectly in-phase or orthogonal due to the presence of frequency split. Therefore, the following is assumed:(8)$$x=A_{x}\cos\left(\omega_{x}t\right)$$
(9)$$y=A_{y}\cos\left(\omega_{x}t\right)+A_{y,Q}\sin\left(\omega_{x}t\right)$$
where *A_y_* and *A_y_*_,*Q*_ are the displacement amplitudes orthogonal and in-phase with the actuation force at the sensing axis, respectively. By substituting Equations (8) and (9) into (4) and (5), we obtain the following:(10)$$\begin{array}{cc} {\dot{A}}_{y} & =-\frac{\omega_{y}}{2Q_{y}}A_{y}+\Delta \omega A_{y,Q}-nA_{g}\Omega A_{y}-\frac{F_{s}}{2m\omega_{x}}\\ & +\left[\frac{\omega_{y}}{2Q_{y}}A_{y}-\Delta \omega A_{y,Q}+nA_{g}\Omega A_{x}+\frac{F_{s}}{2m\omega_{x}}\right]\cos\left(2\omega_{x}t\right)\\ & +\left[\frac{\omega_{y}}{2Q_{y}}A_{y,Q}-\Delta \omega A_{y}\right]\sin\left(2\omega_{x}t\right) \end{array}$$
(11)$$\begin{array}{cc} {\dot{A}}_{y,Q} & =-\frac{\omega_{y}}{2Q_{y}}A_{y,Q}-\Delta \omega A_{y}+\left[\frac{\omega_{y}}{2Q_{y}}A_{y,Q}+\Delta \omega A_{y}\right]\cos\left(2\omega_{x}t\right)\\ & +\left[\frac{\omega_{y}}{2Q_{y}}A_{y}-\Delta \omega A_{y,Q}+nA_{g}\Omega A_{x}+\frac{F_{s}}{2m\omega_{x}}\right]\sin\left(2\omega_{x}t\right) \end{array}$$

By using the stochastic averaging method, the differential equation for the slow signal (amplitude signal) can be obtained:(12)$${\dot{A}}_{y}=-\frac{\omega_{y}}{2Q_{y}}A_{y}+\Delta \omega A_{y,Q}-\Omega A_{y}-\frac{F_{s}}{2m\omega_{x}}$$
(13)$${\dot{A}}_{y,Q}=-\frac{\omega_{y}}{2Q_{y}}A_{y,Q}-\Delta \omega A_{y}$$

By using the Laplace transform on Equations (12) and (13), we obtain the following:(14)$$sA_{y}\left(s\right)=-\frac{\omega_{y}}{2Q_{y}}A_{y}\left(s\right)+\Delta \omega A_{y,Q}\left(s\right)-A_{x}nA_{g}\Omega \left(s\right)-\frac{F_{s}\left(s\right)}{2m\omega_{x}}$$
(15)$$sA_{y,Q}\left(s\right)=-\frac{\omega_{y}}{2Q_{y}}A_{y,Q}\left(s\right)-\Delta \omega A_{y}\left(s\right)$$

Both the Coriolis force 2*mω_x_A_x_*Ω(*s*) and *F_s_* are the external forces in the system. Therefore, assuming that the total force towards the sensing axis is *F_y_*(*s*) = 2*mnA_g_ω_x_A_x_*Ω(*s*) + *F_s_*(*s*) and deducing (14) and (15), we obtain the following:(16)$$H_{Fy}\left(s\right)=\frac{A_{y}\left(s\right)}{F_{y}\left(s\right)}=\frac{s+\frac{\omega_{y}}{2Q_{y}}}{2m\omega_{x}\left[{\left(s+\frac{\omega_{y}}{2Q_{y}}\right)}^{2}+\Delta \omega^{2}\right]}$$
(17)$$H_{Fy,Q}\left(s\right)=\frac{A_{y,Q}\left(s\right)}{F_{y}\left(s\right)}=\frac{-\frac{\omega_{y}}{2Q_{y}}}{2m\omega_{x}\left[{\left(s+\frac{\omega_{y}}{2Q_{y}}\right)}^{2}+\Delta \omega^{2}\right]}$$

Note that the effect of the transfer function *H_Fy_*_,*Q*_(*s*) will be filtered out by the demodulator and low-pass filter in the rate loop; only the effect of (16) is taken into account when modeling system-level noise.

### 3.2. Noise Analysis in the System

The system noise model of the MEMS DRG is shown in Figure 3 based on the dynamic model of the slow signal of the resonator. In Figure 3, *k_vi_* and *k_VF_* are the conversion factors from resonator velocity to sensing current and from actuation voltage to force, respectively; *C_i_* is the integrated capacitance of the charge sensing amplifier (CSA) and *k_m_* is the gain of the modulator. The transfer functions of the PI controller and low-pass filter (LPF) are as follows:(18)$$PI\left(s\right)=K_{p}+\frac{K_{I}}{s}$$
(19)$$LPF\left(s\right)=\frac{\omega_{p}}{s+\omega_{p}}$$
where *K_p_* and *K_i_* are the proportional and integral terms of the PI controller, respectively, and *ω_p_* is the cutoff frequency of the LPF. Since the loop gain is infinite at low frequency due to the PI controller, the Coriolis force is equal to the feedback force in the closed loop:(20)$$2mn\omega_{x}A_{g}A_{x}\Omega =V_{out}k_{m}k_{VF}$$

Therefore, the scale factor of the MEMS DRG system is as follows:(21)$$SF=\frac{V_{out}}{\Omega }=\frac{2mn\omega_{x}A_{g}A_{x}}{k_{m}k_{VF}}$$

The main noise sources in the rate loop are CSA noise at resonant frequency ${\bar{I}}_{wn,cv}$, 1/f noise of the PI amplifier ${\bar{V}}_{PI,fn}$, mechanical noise ${\bar{N}}_{MEMS}$ and electrical noise introduced by the feedback element ${\bar{V}}_{FB}$. The feedback element refers to the circuits after gyroscope output, such as a DAC in a digital circuit and a multiplier in conventional analog circuits.

According to the system-level noise model in Figure 3, the output noise voltage can be obtained:(22)$$V_{out,n}\left(s\right)=\frac{{\bar{I}}_{cv,n}+C_{i}\omega_{x}{\bar{V}}_{PI,fn}}{\omega_{x}H_{Fy}\left(s\right)\left(k_{VF}k_{vi}k_{m}+\frac{C_{i}}{LPF\left(s\right)PI\left(s\right)H_{Fy}\left(s\right)}\right)}+\frac{{\bar{N}}_{MEMS}}{k_{VF}k_{m}}+{\bar{V}}_{FB}$$

The *LPF*(s)*PI*(s) is much larger than 1 within bandwidth. By combining Equation (21), the output noise of the system can be given as follows:(23)$$\Omega_{n}\left(s\right)=\frac{{\left(s+\frac{\omega_{y}}{2Q_{y}}\right)}^{2}+\Delta \omega^{2}}{nA_{g}A_{x}k_{vi}\omega_{x}\left(s+\frac{\omega_{y}}{2Q_{y}}\right)}\left({\bar{I}}_{cv,n}+C_{i}\omega_{x}{\bar{V}}_{PI,fn}\right)+\frac{{\bar{N}}_{MEMS}+k_{m}k_{VF}{\bar{V}}_{FB}}{2mn\omega_{x}A_{g}A_{x}}$$

From Equation (23), one can see that ${\bar{I}}_{wn,cv}$ and ${\bar{V}}_{PI,fn}$ are suppressed by the transfer function of the mechanical structure, but the frequency split Δ*ω* may deteriorate the shaping ability of the resonator at low frequencies. ${\bar{V}}_{elec}$ and ${\bar{N}}_{MEMS}$ may directly affect the output of the DRG and deteriorate the noise floor. Since the MEMS DRG possesses a very high Q factor as Section 2 discussed, the effect of ${\bar{N}}_{MEMS}$ is negligible compared with other electrical noises [14].

The simulation verification of the system noise model is given in Figure 4. The circuit system shown in Figure 2 is built in the Cadence IC environment, where the LRC electrical equivalent is used for the sensing element in order to perform co-simulation [19].

Figure 4a gives the output noise spectral density for different frequency split cases. Only the transient noise of the CSA is added in the setup to avoid interference from other noise sources. The black, red and blue lines are the output noise spectra with frequency splits of 0 Hz, 0.5 Hz and 1 Hz, respectively. The simulation results show that the noise floor increases at low frequency with the increase in frequency split, which is consistent with Equation (23).

Figure 4b shows the verification of the effect of ${\bar{V}}_{FB}$. The blue line is the output spectrum with only CSA noise added, the red line is the spectrum when ${\bar{V}}_{FB}=5\;\mu V/\sqrt{\mathrm{Hz}}$ is artificially injected and the black line is the spectrum with both included. From the simulation results, it is clear that ${\bar{V}}_{FB}$ directly affects the output and deteriorates the noise floor, which coincides with the analysis above.

Figure 4c shows the verification of the effect of 1/f noise of the PI amplifier. The red and blue lines show the output noise spectra with and without the addition of PI amplifier noise, respectively. Since the 1/f noise of the PI amplifier is higher at a low frequency [20], the output noise floor is raised.

### 3.3. Angle Random Walk Analysis and Optimization

The above analysis gives the relationship between the output noise and frequency. In order to optimize the performance of the MEMS DRG, the ARW needs to be estimated from the noise spectrum. The relationship between Allan variance and the power spectrum density is given by the following [21]:(24)$$\sigma^{2}\left(\tau \right)=4\int_{0}^{\infty }\Omega_{n}^{2}\left(f\right)\frac{\sin^{4}\left(\pi f\tau \right)}{{\left(\pi f\tau \right)}^{2}}df$$

The value of ARW can be obtained by reading the slope line at *τ* = 1 [21]:(25)$$ARW=\sigma \left(1\right)=2\sqrt{\int_{0}^{\infty }\Omega_{n}^{2}\left(f\right)\frac{\sin^{4}\left(\pi f\right)}{{\left(\pi f\right)}^{2}}df}$$

The main lobe of the window function sin^2^(*πf*)/(*πf*) is within 1 Hz, while the side lobes are attenuated. Therefore, a low ARW design needs to focus on the optimization of output noise within 1 Hz. The following optimization methods ensure the low ARW.

Firstly, the selected MEMS DRG possesses a very high Q factor, with its own ARW contribution being less than 0.001°/√h [14]. Secondly, compared with mode-split gyroscopes, a mode-matched gyroscope has higher mechanical sensitivity, and the noise contribution from the front end (CSA) is suppressed by the transfer function of the mechanical structure. Thirdly, the low-frequency noise of the PI amplifier can be converted to output noise, and this issue should be taken into account when designing the integrated circuit. A low ripple chopping technique is used here to reduce this part of the noise. Fourthly, switching to phase-sensitive modulation instead of the traditional analog multiplier is used here, which greatly reduces the noise contribution in the modulation feedback stage. Lastly, the manual frequency method is used to reduce the frequency split, and the specific implementation can be referred to in previous work [14,22]. Benefiting from a highly symmetrical mechanical structure as described in Section 2, the frequency split Δω varies slightly as the environment changes. The implementation of the low-noise interface circuit will be described in the next section.

## 4. Circuit Implementation Details

### 4.1. Current Sensing Amplifier

The transistor level circuit of the CSA in this paper is shown in Figure 5. The amplifier employs a three-stage topology for high DC gain. The first two stages employ a fully differential input to reduce system offset. The amplifier employs trans-conductance with a capacitance feedback compensation (TCFC) topology in order to maintain stability and lower the power dissipation [23].

In Figure 5, *C_p_* is the total parasitic capacitance of the front end, including the parasitic capacitance of the sensing element and the parasitic capacitance of the CSA itself. The noise current ${\bar{I}}_{wn,cv}$ in Figure 3 is the ratio of the equivalent input voltage noise of the CSA ${\bar{V}}_{n,cv}$ to the capacitance *C_p_*:(26)$${\bar{I}}_{wn,cv}=\omega_{x}C_{p}{\bar{V}}_{n,cv}$$

Therefore, the main focus of the CSA design is to reduce the equivalent input voltage noise at the resonant frequency. The resonant frequency of the MEMS DRG used in the paper is about 4.65 kHz, at which point the 1/f noise is still large for a traditional CMOS amplifier. Therefore, the circuit design should focus on optimizing the 1/f noise at the resonant frequency. Since the equivalent input voltage noise of the second and third stages will be suppressed by the gain of the front stage, the first stage noise contribution is the main focus of our concern. A simple current mirror structure is used in the first stage to avoid the introduction of more noise sources. Since the 1/f noise coefficient of the PMOS transistor is smaller than that of the NMOS transistor, the input transistor (M_2_) uses the PMOS transistor to reduce the 1/f noise. The area of the current mirror transistors (M_3a_, M_3b_) should be designed to be large enough to suppress the 1/f noise by themselves. In addition, the area of M_2_ should not be too large so as to avoid excessive parasitic capacitance *C_p_*, which should be optimized in compromise with the 1/f noise. The simulation result of the voltage noise of the CSA is given in Figure 6. After optimization, the equivalent input voltage noise density at 4.65 kHz is about 7.43 nV/√Hz.

### 4.2. Amplifier in PI Controller

The PI and LPF circuits used in the paper are shown in Figure 7, both of which are completed by only one amplifier. From the discussion in Section 3, it can be seen that the 1/f noise of the amplifier in the PI controller needs to be reduced in order to lower the output noise. Therefore, a low 1/f noise amplifier is proposed here, as shown in Figure 8. The main structure of the amplifier is the same as the CSA in Figure 5. The circuit adopts the chopping technique to reduce the low-frequency noise. At the same time, a continuous-time AC-coupled ripple reduction loop (RRL) is added to the amplifier to eliminate the high-frequency ripple caused by chopping [24]. Figure 9a gives the spectral density of the equivalent input voltage noise before and after chopping, and the noise at 1 Hz decreases from 1.7 μV/√Hz to 11.5 nV/√Hz after chopping. Figure 9b shows the verification of the ripple suppression loop, where the output ripple drops from 500 mV to 100 μV in 400 μs.

### 4.3. Feedback Element

The noise from the feedback element ${\bar{V}}_{FB}$ may directly affect the output noise floor, as can be seen from the analysis in Section 3.2. Moreover, Equation (21) also shows that the variation of the gain of the feedback element *k_m_* also affects the scale factor of the gyroscope system. A conventional analog closed-loop scheme based on discrete devices employs multipliers for force feedback [14]. However, an analog multiplier with low noise, high linearity and good full temperature characteristics is difficult to design in integrated circuits since it requires complex compensation techniques [25]. Although the digital–analog hybrid structure multiplier can achieve a better performance, it includes an ADC, digital signal processing (DSP) and DAC itself [26], which does not facilitate a single-chip integrated design.

A phase-sensitive modulator is used here instead of a multiplier to avoid the above problems, as Figure 10 shows. Although high-order harmonic components of the resonant frequency exist at the output of the modulator, these harmonic components can be filtered by a mechanical resonator with a high Q factor. In Figure 10, *V_demout_* represents the outputs of the demodulator, *V_PIout_*_±_ represents the outputs of the PI controller in the rate control loop and *clk_v* is the clock in phase with velocity. The modulation feedback circuit contains only switches and is easy to integrate. Since the resistance is only about 100 ohms when the switch turns on, the thermal noise introduced by the switch is only 1.29 nV/√Hz according to the thermal noise equation ${\bar{V}}_{R}=\sqrt{4k_{B}TR}$, which is much smaller than the effect of other noise sources.

### 4.4. Other Circuit Elements

In addition to the above units, the MEMS DRG interface ASIC also includes a self-clocking circuit, a VGA (non-linear multiplier) and a ΣΔ ADC.

The self-clocking circuit is shown in Figure 11. The input of the self-clocking circuit is connected to the output of the TIA in the drive loop, and the resonator velocity signal is locked, multiplied and 90° phase-shifted by the PLL. The 90° phase shift is generated by exclusive OR (XOR) operation between *clk_v* and its multiple frequency clock.

The design details of the low harmonic distortion nonlinear multiplier and the ΣΔ ADC can be found in our previous works [27,28].

## 5. Experimental Results and Discussion

The interface integrated circuit for the MEMS DRG is implemented using a 0.35 μm CMOS process. The chip photograph with the drive loop, rate and quadrature loop, self-clocking circuit, low-pass ΣΔ modulator and digital circuits is shown in Figure 12a. The total area of the ASIC is 4.5 mm × 4.3 mm, and the power dissipation is 50 mW, with a ±2.5 V supply voltage. The MEMS DRG is sealed in a vacuum package and connected to the ASIC on a print circuit board (PCB). The test board is shown in Figure 12b.

Using an Agilent 35670A dynamic signal analyzer to analyze the analog output noise of the MEMS DRG and setting the resolution to 31.25 mHz, the output noise spectrum is tested in Figure 13. The spectrum shows that the output noise is well suppressed at low frequencies, which is consistent with the analysis in Section 3. In addition, it can be seen that the noise starts to rise at 78.1 mHz, which indicates that the main source of output noise at this frequency is still the noise of the CSA. Since the frequency split Δω may deteriorate the shaping ability of the resonator before Δω according to Equation (23), the noise spectrum also demonstrates that the Δ*ω* is below 78.1 mHz. This result also demonstrates that the output noise introduced by the feedback element using the phase-sensitive modulator is negligible, which is consistent with the analysis in Section 4.3. The noise shaping ability at high frequencies deteriorates due to the limited bandwidth of the MEMS DRG system. The system bandwidth can therefore be estimated from the frequency spectrum, which is about 15 Hz. The scale factor of the analog output is 40 mV/°/s, and thus, the noise floor at low frequencies is about 7.5 μ°/s (at 31.25 mHz resolution), which is similar to the simulation results in Section 3.2.

The digital outputs of the MEMS DRG under rotations of −50°/s, −25°/s, −12.5°/s, −5°/s, −2.5°/s, −1.25°/s, −0.5°/s, −0.25°/s, −0.1°/s, −0.05°/s, 0.05°/s, 0.1°/s, 0.25°/s, 0.5°/s, 1.25°/s, 2.5°/s, 5°/s, 12.5°/s, 25°/s and 50°/s were measured to characterize the scale factor and the nonlinearity of the system, and the test results are given in Figure 14a,b. A scale factor of 116,850 LSB/(°/s) over the ±50°/s full range and a maximum nonlinearity of 320 ppm were obtained according to the measurement.

The zero-rate output (ZRO) of the MEMS DRG was tested at room temperature with a sample frequency of 2 Hz. The gyroscope output was recorded for 4 h, as shown in Figure 15a,b. The bias stability (1σ, at a binning time of 10 s), bias instability and ARW of the gyroscope system were 0.47°/h, 0.038°/h and 0.0075°/√h, respectively.

Table 2 lists a comparison of the performances of DRG systems in recent years. The front-end and control circuits in this paper are both implemented by the ASIC, offering a higher level of integration than in the comparative literature. The study [4] shows the laboratory performance of the Boeing MEMS DRG by using the dSPACE semi-physical simulation platform, which demonstrates that the MEMS DRG possesses a navigation-level performance. The studies [15,22] used a lock-in amplifier HF2LI from Zurich Instruments to control the disc gyro in order to measure its ultimate performance. Compared with [14,15,22,29,30], this paper has advantages in terms of ARW and BI.

## 6. Conclusions

This paper presents an interface ASIC for an MEMS DRG. The ASIC incorporates a drive loop, a rate loop, a quadrature loop, a self-clocking circuit, a low-pass ΣΔ modulator and digital circuits with a die area of 4.3 mm × 4.5 mm. The sources of the output noises were analyzed to give guidelines for circuit design based on the slow signal equivalence of the gyroscope dynamics. By using an integrated circuit to optimize the noise of key elements, a performance of 0.0075°/√h ARW and 0.038°/h BI was achieved. This ASIC is also suitable for other types of mode-matched gyroscopes such as a quad mass gyroscope, a dual foucault pendulum gyroscope, etc.

## Figures and Tables

**Figure 1 micromachines-14-01256-f001:**
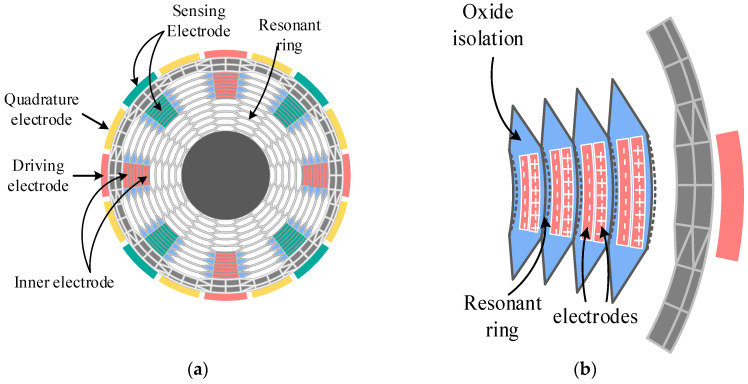
(**a**) Top view of the MEMS DRG. (**b**) Top view of radial electrodes.

**Figure 2 micromachines-14-01256-f002:**
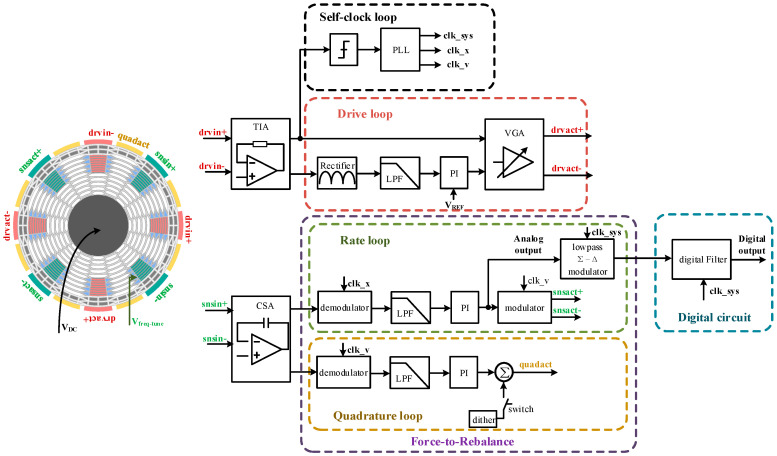
Block diagram of the DRG system.

**Figure 3 micromachines-14-01256-f003:**
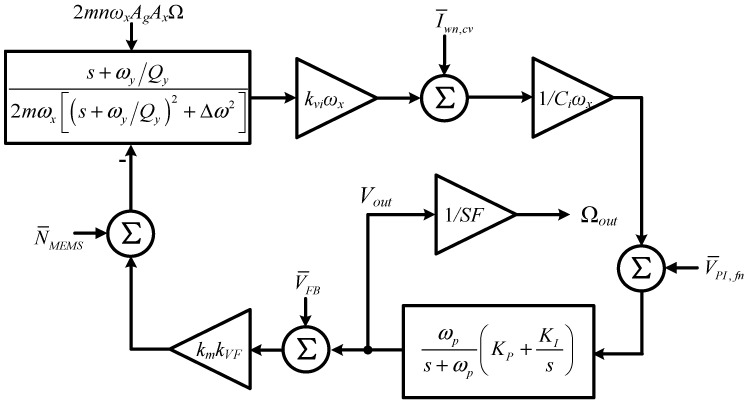
System noise model of the MEMS DRG.

**Figure 4 micromachines-14-01256-f004:**
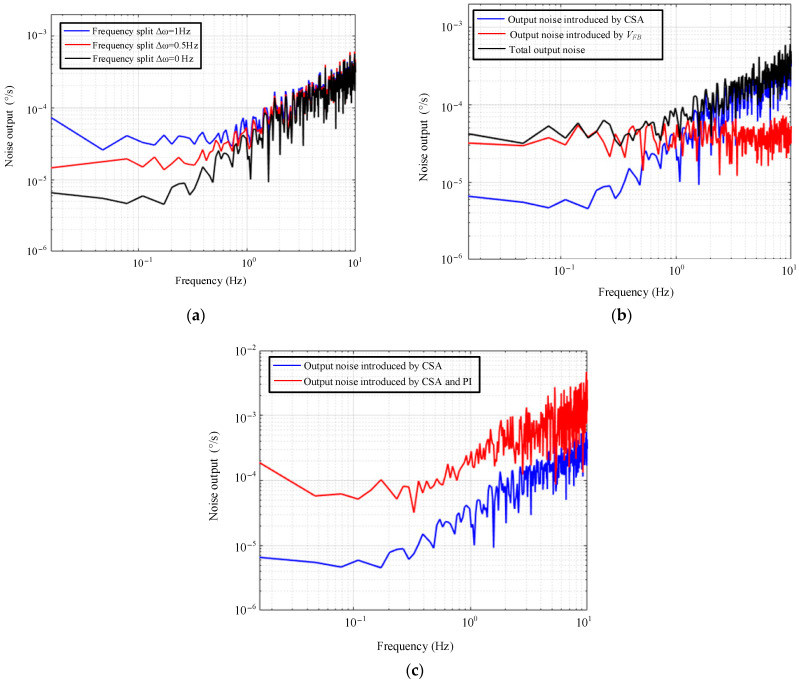
Simulation verification of (**a**) frequency split, (**b**) influence of feedback element and (**c**) 1/f noise of PI amplifier.

**Figure 5 micromachines-14-01256-f005:**
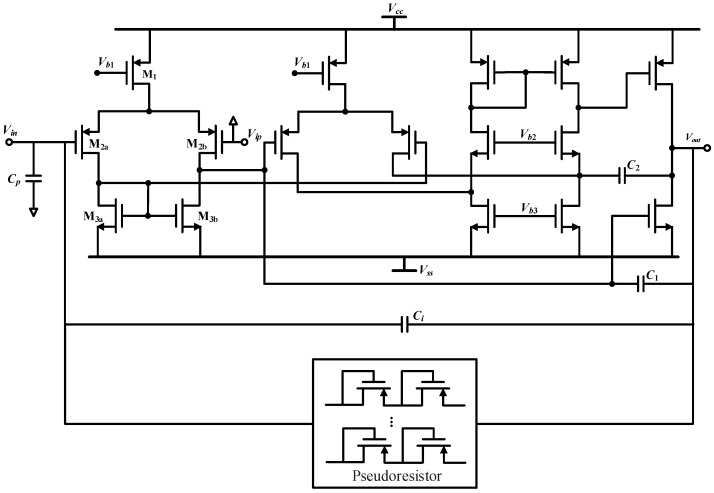
The transistor level circuit of the CSA.

**Figure 6 micromachines-14-01256-f006:**
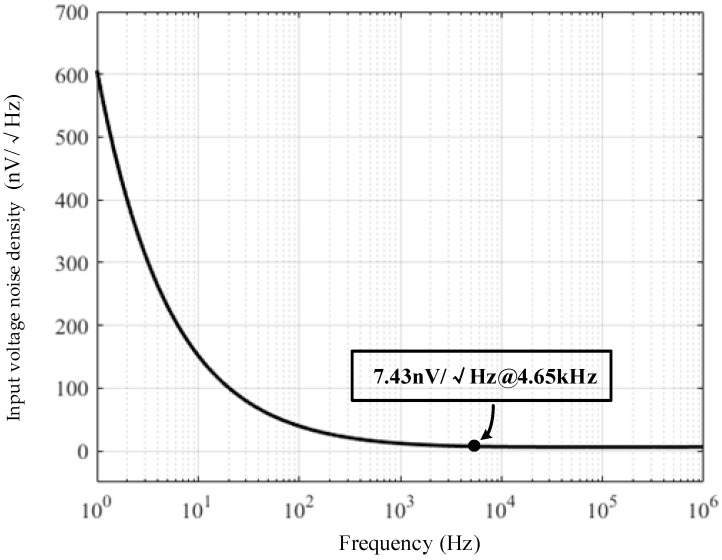
Input voltage noise density of the CSA.

**Figure 7 micromachines-14-01256-f007:**
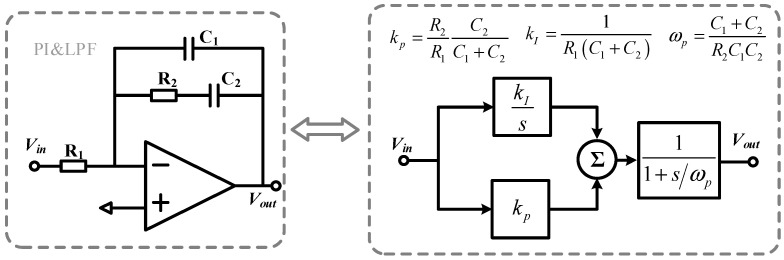
Circuit implementation of PI and LPF.

**Figure 8 micromachines-14-01256-f008:**
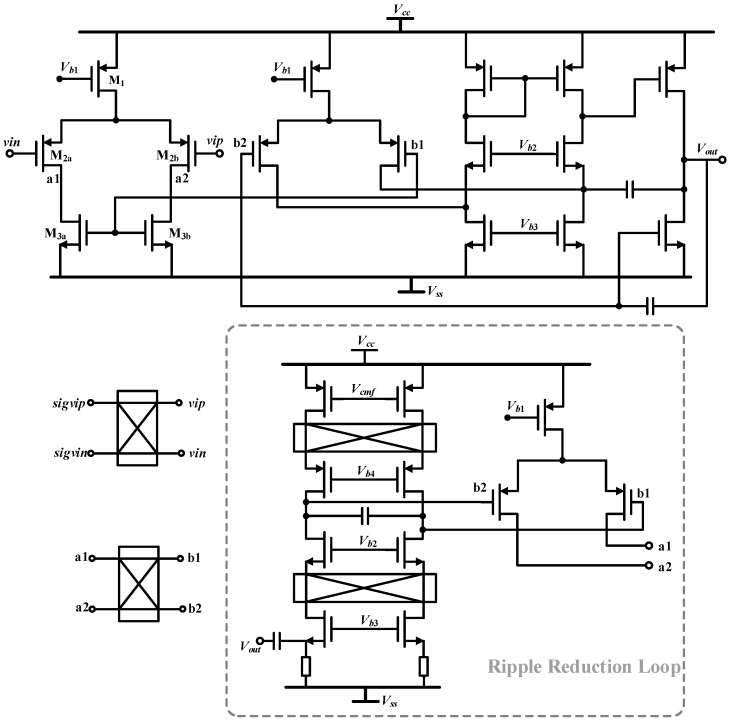
The transistor circuit of the low 1/f noise amplifier in PI and LPF.

**Figure 9 micromachines-14-01256-f009:**
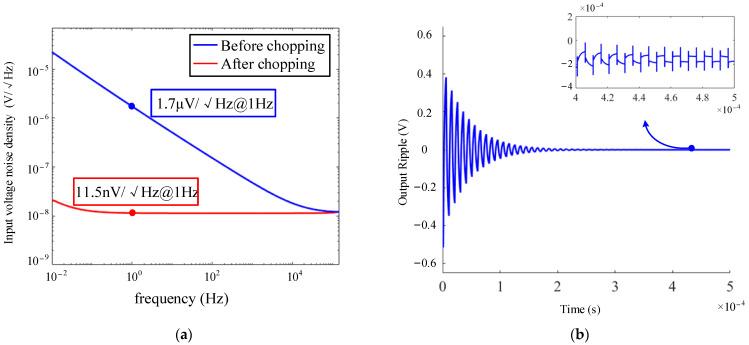
Simulation of the low 1/f noise amplifier with (**a**) input voltage noise density with and without the chopping technique, and (**b**) verification of the ripple suppression loop.

**Figure 10 micromachines-14-01256-f010:**
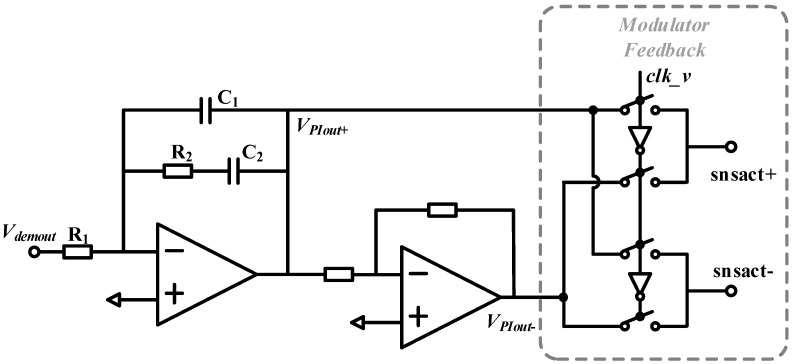
Phase-sensitive modulation feedback scheme.

**Figure 11 micromachines-14-01256-f011:**
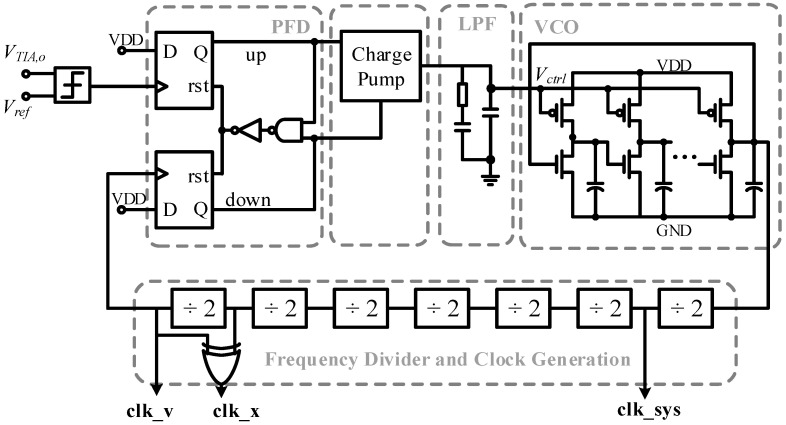
The self-clocking circuit in the ASIC.

**Figure 12 micromachines-14-01256-f012:**
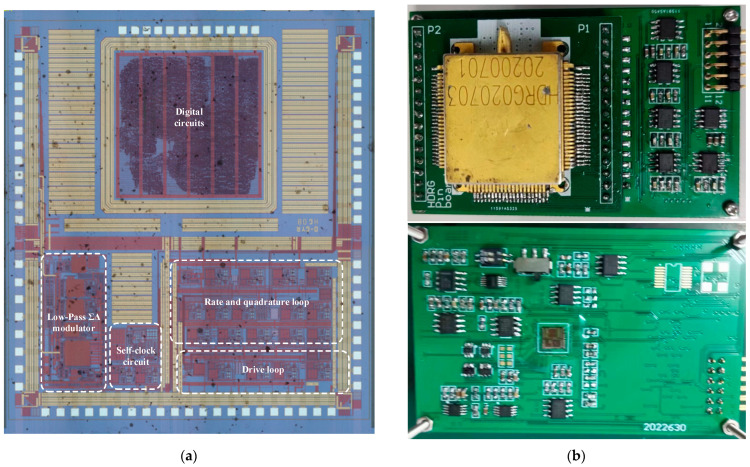
(**a**) ASIC photograph and (**b**) MEMS DRG test board.

**Figure 13 micromachines-14-01256-f013:**
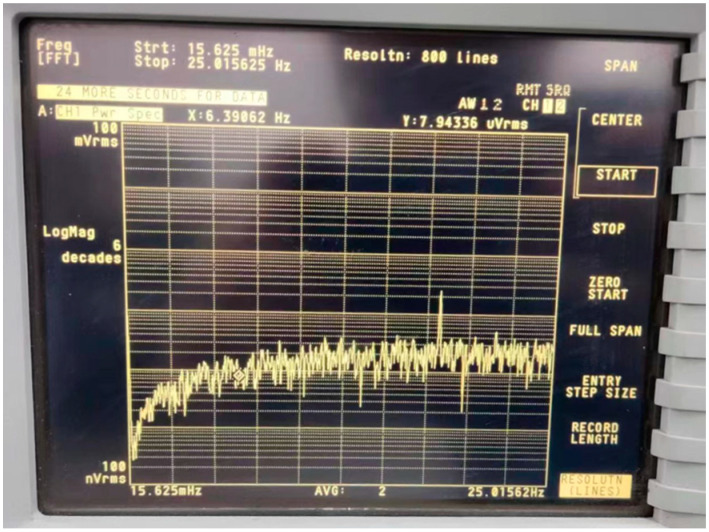
Analog output noise analysis using Agilent 35670A.

**Figure 14 micromachines-14-01256-f014:**
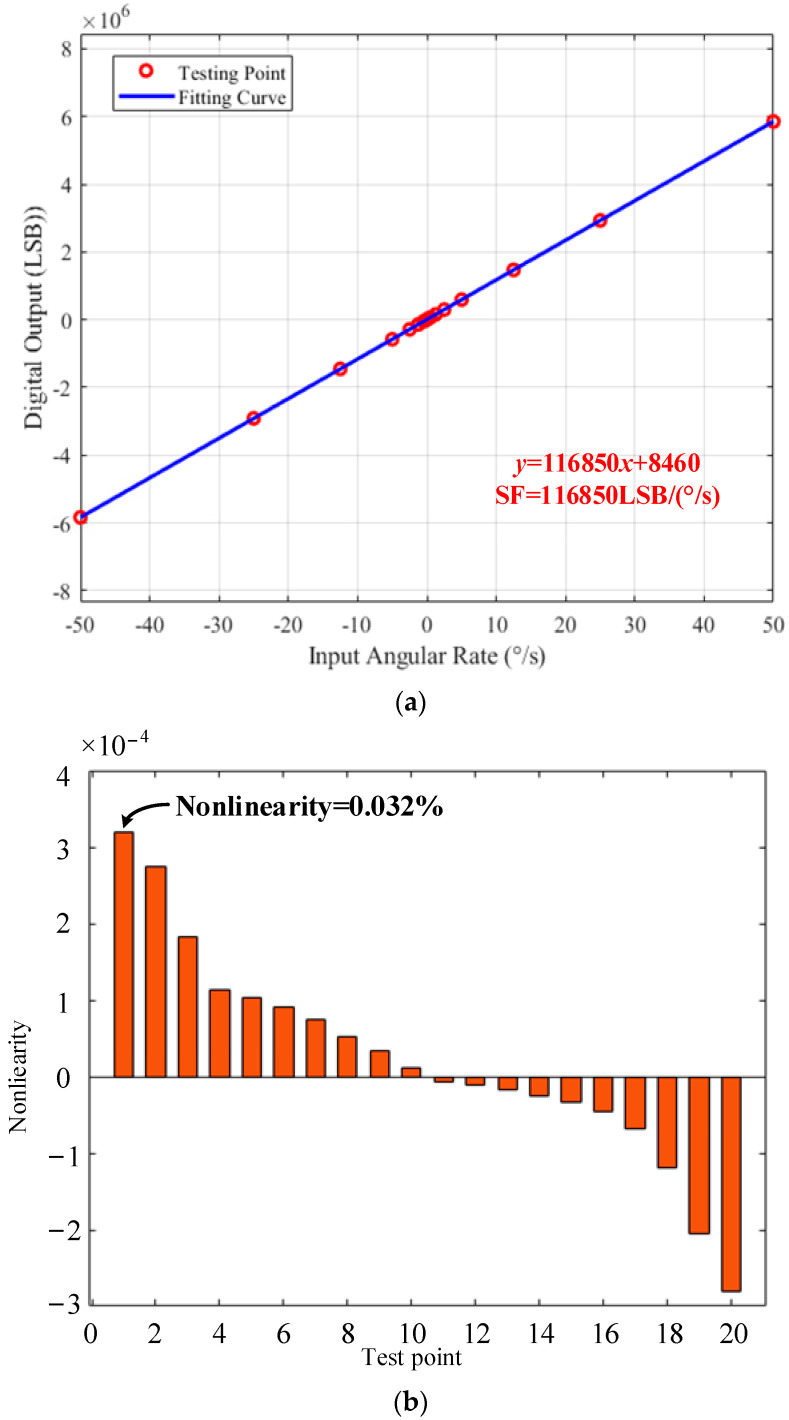
Scale factor tests over ±50°/s full range. (**a**) Scale factor test and linear curve fitting over full range. (**b**) Scale factor nonlinear curve and calculated maximum nonlinearity in the tested dynamic range.

**Figure 15 micromachines-14-01256-f015:**
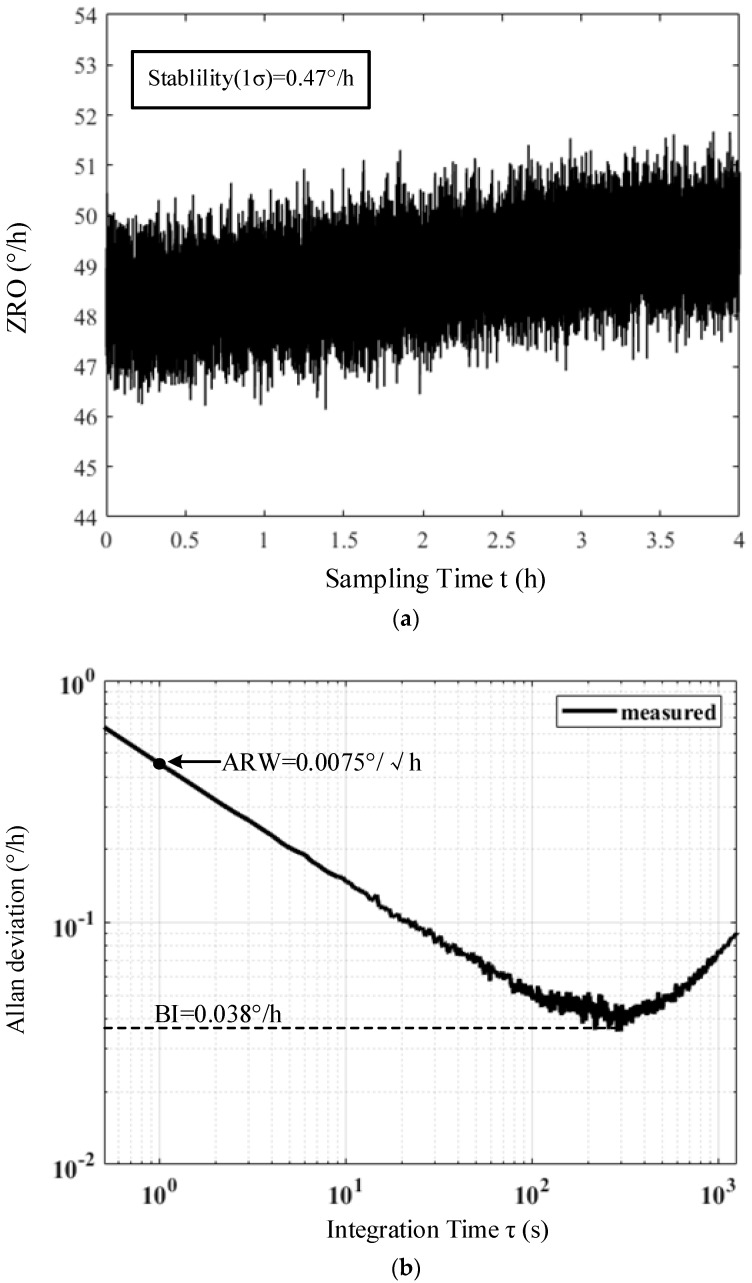
(**a**) Origin data of ZRO at room temperature and (**b**) Allan deviation of the origin data.

**Table 1 micromachines-14-01256-t001:** Key parameters of MEMS DRG.

Parameters	Values	Unit
Resonator diameter	8	mm
Beam width	12	μm
Gap width (*d*)	11	μm
Effective mass (*m*)	2.54 × 10^−6^	kg
Stiffness coefficient (*k*)	2168.2	N/m
Damping coefficient (*D*)	1.769 × 10^−7^	N/(m/s)
Resonant frequency (*f*)	4650	Hz
Quality factor (*Q*)	5.1 × 10^5^	-
Mechanical bandwidth (*ω_m_*)	0.0349	rad/s
Oscillation amplitude (*A_x_*)	4	μm

**Table 2 micromachines-14-01256-t002:** Performance comparison of MEMS DRGs.

Year	Ref	Circuit Type	ARW °/√h	BI (°/h)	Full Scale (°/s)
2014	[4]	dSPACE platform	0.003	0.01	-
2014	[27]	PCB and CMOS front-end	0.48	20	-
2016	[20]	HF2LI	-	4	-
2018	[14]	Analog discrete circuit	0.01	0.04	±100
2020	[15]	HF2LI	0.018	0.23	±20
2022	[28]	PCB front-end and digital ASIC	0.05	0.42	±300
2023	This paper	ASIC	0.0075	0.038	±50

## Data Availability

The data that support the findings of this study are available from the corresponding author upon reasonable request.

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
