# Peer review of "A Low-Noise Interface ASIC for MEMS Disk Resonator Gyroscope"

_micromachines, 2023, doi:10.3390/mi14061256_

Round 1

Reviewer 1 Report

This manuscript presents the design, optimization, and characterization of an interface ASIC for MEMS disk resonator gyros. The topic is interesting and important to the field. The manuscript is well organized and the results demonstrated are impressive. However, a few minor issues need to be clarified:

1. Is the DC polarization voltage provided by the ASIC or applied off-chip? HV charge pump is usually an important block in MEMS resonator gyro interface ASIC. Its noise performance and power consumption are important factors to consider in gyro ASIC design.

2. What is the voltage level needed for “quadact” to cover the quadrature tuning range across process variations of the MEMS?

3. Figure 2 shows a dither signal in the quadrature loop. What is the purpose and the operation principle of this signal?

4. What is the impact of phase delays in the amplifiers? Is there a phase tuning mechanism in the drive loop to ensure locking of oscillation to the resonance peak frequency of the drive mode? Similarly, can the sense demodulation phase be adjusted to ensure proper projection of in-phase and quadrature signals?

5. The noise analysis considers the CSA noise only at the resonant frequency. Could there be flicker noise coupling to carrier frequency due to transducer nonlinearity?

6. What is the closed-loop bandwidth of the gyro with FTR operation?

English language quality is okay. Only minor editing and corrections required.

Reviewer 2 Report

This paper presents an interface ASIC for MEMS DRG. And a good performance is achieved using the proposed noise optimization method. The experiment is also carried out.  I think the paper is well organized and presented. 

Author Response

Thank you very much for your constructive and insightful comments on our manuscript and we want to express our sincere appreciation to you. Your comments on the manuscript have been carefully considered and the revised manuscript has been modified accordingly.

Reviewer 3 Report

This paper presents an ASIC for MEMS DRG and a nice result of the gyro performance. Although the authors highlighted their test low ARW and BI, this wok did not clearly illustrate how to reach this performance and give the valuable analysis about the circuit design but in an genral vague way (see comments on fig.4 and conclusions in p9). In other wors, we can not find any estimation or simulation about the overall noise  by combining all the noise sources to obtain a performance of 0.0075°/√h ARW and 0.038°/h BI. Moreover, we can not find any optimization method for the circuit since the basic modules adopted by the work seems to be conventional ordinary designs. The key important parameters that affect the gyro performance such as quadrature coupling and the frequency split are not ananlyzed, which degrades the credibility of the article. 

Besides, I have some questions or suggestions:

1. The noise optimization method should be clearly demonstrated especially the optimization target for the BI and ARW.

2. The quadrature coupling signal should be given after adopting quadrature loop. The authors should give some comments on the quadrature effect in DRG.

3. P9, the authors mentioned that the manual frequency trimming method proposed by ADI is used in this paper. It seems that 'the manual frequency trimming' is confusing. Does ADI use the manual trimming in [20]?

Morever, what is the final frequency split of the DRG? If the method in [20] is utilized , the noise contribution by the method should be analyzed. 

Some minor writting issuses should be revised.

Round 2

Reviewer 3 Report

 I have no comments on the technique aspects but some english writing issues. Since there are some trivial points, I did not mention them one by one. I think the authors should recheck the whole manuscript again. 

 I have no comments on the technique aspects but some english writing issues. Since there are some trivial points, I did not mention them one by one. I think the authors should recheck the whole manuscript again.